# Resolving drug selection and migration in an inbred South American *Plasmodium falciparum* population with identity-by-descent analysis

Manuela Carrasquilla[1,2☉]*, Angela M. Early[1,2☉]*, Aimee R. Taylor[2,3], Angélica Knudson Ospina[4], Diego F. Echeverry[5,6], Timothy J. C. Anderson[7], Elvira Mancilla[8], Samanda Aponte[9], Pablo Cárdenas[10], Caroline O. Buckee[3], Julian C. Rayner[11,12], Fabián E. Sáenz[13], Daniel E. Neafsey[1,2]*, Vladimir Corredor[9]*

1 Department of Immunology and Infectious Diseases, Harvard T.H.Chan School of Public Health, Boston, Massachusetts, United States of America, 2 Infectious Disease and Microbiome Program, Broad Institute of MIT and Harvard, Cambridge, Massachusetts, United States of America, 3 Center for Communicable Disease Dynamics, Harvard T.H.Chan School of Public Health, Boston, Massachusetts, United States of America, 4 Departamento de Microbiología, Facultad de Medicina, Universidad Nacional de Colombia, Bogotá, Colombia, 5 Departamento de Microbiología, Facultad de Salud, Universidad del Valle, Cali, Colombia, 6 Centro Internacional de Entrenamiento e Investigaciones Médicas (CIDEIM), Cali, Colombia, 7 Program in Disease Intervention and Prevention, Texas Biomedical Research Institution, San Antonio, Texas, United States of America, 8 Secretaría Departamental de Salud del Cauca, Popayán, Colombia, 9 Departamento de Salud Pública, Facultad de Medicina, Universidad Nacional de Colombia, Bogotá, Colombia, 10 Department of Biological Engineering, Massachusetts Institute of Technology, Cambridge, Massachusetts, United States of America, 11 Wellcome Sanger Institute, Hinxton, Cambridge, United Kingdom, 12 Cambridge Institute for Medical Research, University of Cambridge, Cambridge, United Kingdom, 13 Centro de Investigación para la Salud en América Latina, Facultad de Ciencias Exactas y Naturales, Pontificia Universidad Católica del Ecuador, Quito, Ecuador

☉ These authors contributed equally to this work.
* carrasquilla@mpiib-berlin.mpg.de (MC); early@broadinstitute.org (AME); neafsey@broadinstitute.org (DEN); vcorredore@unal.edu.co (VC)

## Abstract

The human malaria parasite *Plasmodium falciparum* is globally widespread, but its prevalence varies significantly between and even within countries. Most population genetic studies in *P. falciparum* focus on regions of high transmission where parasite populations are large and genetically diverse, such as sub-Saharan Africa. Understanding population dynamics in low transmission settings, however, is of particular importance as these are often where drug resistance first evolves. Here, we use the Pacific Coast of Colombia and Ecuador as a model for understanding the population structure and evolution of *Plasmodium* parasites in small populations harboring less genetic diversity. The combination of low transmission and a high proportion of monoclonal infections means there are few outcrossing events and clonal lineages persist for long periods of time. Yet despite this, the population is evolutionarily labile and has successfully adapted to changes in drug regime. Using newly sequenced whole genomes, we measure relatedness between 166 parasites, calculated as identity by descent (IBD), and find 17 distinct but highly related clonal lineages, six of which have persisted in the region for at least a decade. This inbred population structure is captured in more detail with IBD than with other common population structure analyses like

**Data Availability Statement:** All relevant data are within the manuscript, its Supporting information files, or available on public servers. Whole genome sequence data for Ecuador samples are available on the Sequence Read Archive as BioProject PRJNA759192. Whole Genome sequence data from Colombia samples are available on the European Nucleotide Archive and accession numbers are provided in S3 Table. Recoded GoldenGate calls from Echeverry, et al 2013 are available in S4 Table.

**Funding:** The work performed in Colombia was supported by the Newton Caldas Fund Institutional Links G1854 Award to JCR and VC. The Medical Faculty at Universidad Nacional de Colombia provided support with awards HERMES 35988 and 32309 to VC. Whole genome sequencing of all samples obtained in Guapi was financially supported by MalariaGEN and the Wellcome Trust (206194, 090770). Financial support for Ecuador was provided by Pontificia Universidad Católica del Ecuador, grants M13416, N13416 and O13087 to FES and Ministerio de Salud Pública del Ecuador. This project has been funded in whole or in part with Federal funds from the National Institute of Allergy and Infectious Diseases (NIAID), National Institutes of Health, Department of Health and Human Services, under Grant Number U19AI110818 to the Broad Institute awarded to DEN. TA is supported by 5R37 AI048071 from NIAID. In all cases, the funders had no role in study design, data collection and analysis, decision to publish, or preparation of the manuscript.

**Competing interests:** The authors have declared that no competing interests exist.

PCA, ADMIXTURE, and distance-based trees. We additionally use patterns of intra-chromosomal IBD and an analysis of haplotypic variation to explore past selection events in the region. Two genes associated with chloroquine resistance, *crt* and *aat1*, show evidence of hard selective sweeps, while selection appears soft and/or incomplete at three other key resistance loci (*dhps*, *mdr1*, and *dhfr*). Overall, this work highlights the strength of IBD analyses for studying parasite population structure and resistance evolution in regions of low transmission, and emphasizes that drug resistance can evolve and spread in small populations, as will occur in any region nearing malaria elimination.

## Author summary

Malaria caused by *Plasmodium falciparum* is a leading cause of mortality in young children, primarily in sub-Saharan Africa. Safe and effective antimalarials have spurred decades of declining prevalence, but drug resistance threatens global elimination efforts. In the Americas, *P. falciparum* transmission and genetic diversity are low. The region accounts for only a small proportion of global infections and mortality, yet it has historically contributed to the emergence and dissemination of *P. falciparum* antimalarial drug resistance. Genomic surveillance in the region can detect emerging drug resistance, dissect its evolutionary dynamics, and impede its spread to highly endemic regions. In this work, we generated and analyzed whole genome sequence data from samples collected in the Pacific Coast region of Colombia and Ecuador, a hotspot of *P. falciparum* malaria. This region exhibits a large proportion of single-clone infections as well as long-term clonal persistence. Our study analyzes this inbred population structure using relatedness estimates calculated as identity-by-descent. We describe the spatial and temporal dynamics of clonal transmission and outcrossing, demonstrating that selection is still effective in this small, inbred population. These findings will support future drug resistance surveillance in regions with intense interventions and declining prevalence.

## Introduction

*Plasmodium falciparum* accounts for the vast majority of malaria mortality globally [1]. High-transmission regions like sub-Saharan Africa bear the greatest mortality, morbidity and economic burden, but malaria caused by *P. falciparum* also imposes significant health and economic burdens in regions of low transmission, including South America [2]. Insights gained in regions of high transmission are often applied to regions of low transmission (and vice versa), however, multiple key biological conditions vary with transmission level including effective population size, outcrossing rates, intrahost competition, and host immunity. We do not fully understand the extent to which variation in these conditions impacts epidemiological, ecological, and evolutionary dynamics under different transmission regimes, or whether these differences necessitate a shift in analysis approaches. This is of growing relevance as more countries experience transmission declines and approach malaria elimination.

The Americas are the global region with the lowest levels of endemic *P. falciparum* transmission and so present a rich opportunity for studying the evolutionary dynamics of small parasite populations. In the Americas, Venezuela accounts for 40% of all *P. falciparum* cases with important pockets of transmission also present in Colombia and Ecuador, with the former accounting for 20% of the regional cases [1]. More than 80% of cases in Colombia occur in the

Pacific Coast Region [3]. After a period of sustained decline in malaria incidence between 2010–2017, there has been a recent increase in malaria in the Americas, reaching more than 900,000 reported cases in 2017 [1]. The rise in cases is consistent with an increase in gold mining activities, an important driver of malaria transmission in the region, particularly for *P. falciparum* [4–7]. Another contributing factor is driven at least in part by instability in Venezuela impacting malaria control; with no imminent solution to the ongoing political and humanitarian crisis there, incidence is likely to remain on the rise, putting at risk elimination efforts not only in Venezuela but in neighboring countries [8,9]. The evolution and spread of drug resistance has the potential to cause further setbacks. Drug resistance has arisen independently in South American *P. falciparum* populations multiple times in the past decades [10–12], and appears to be doing so again, with the recent novel emergence in Guyana of a C580Y mutation in Kelch13 that confers resistance to artemisinin, the current frontline antimalarial [13]. This highlights the importance of continued efforts in genomic surveillance in South America, particularly given the high human mobility currently taking place throughout the region.

The parasite population along the Pacific Coast of Colombia and Ecuador contains a low level of genetic diversity that reflects historical founding events as well as effective malaria control campaigns [14–16]. A large proportion of infections are monoclonal, which results in a low population-level (as opposed to meiotic) recombination rate and permits the long-term persistence of clonal genomes [17–19]. Standard population genetic theory would suggest that these are adverse conditions for adaptive evolution [20], but natural selection still operates effectively, as evidenced by the rapid spread of drug resistance alleles throughout the region in recent decades [21].

Prior genetic studies of parasites in the region have used neutral microsatellites [22] and SNP panels [17,23], which may lack the resolution required to reveal fine-scale differences among samples in this inbred parasite population. We therefore generated new whole-genome sequence data from 207 *P. falciparum* samples collected between 2013 and 2017 from multiple sites along the Pacific Coast of Ecuador and Colombia. Our sampling focused on the municipalities of Santa Bárbara de Iscuandé, Guapi, and Timbiquí in Colombia and on the municipalities of San Lorenzo and Esmeraldas in Ecuador. Relative to the region as a whole, these sites experienced high malaria caseloads—and in some cases local epidemics—during this time period [4,18,24,25]. To better understand population structure and selection dynamics in the region, we use a relatedness framework based on identity by descent (IBD) and compare these results to other common analytic approaches. Whole-genome IBD measurements confirm the presence of long-persisting clonal lineages across the region and more fully describe the high level of inbreeding that characterizes this population. In addition, we use IBD signatures to characterize instances of strong selection in both known and novel regions of the genome. Overall, we show that sufficient recombination has occurred in this population to enable hard and soft selective sweeps, which have responded to both the initiation and removal of drug pressure. This provides strong genomic evidence that *Plasmodium* populations, even when small and isolated, will not remain static as they approach elimination but may continue to adapt successfully to human interventions.

## Results

### Whole-genome sequencing finds predominantly monoclonal infections along the Ecuador-Colombia Pacific Coast Region

We performed whole genome sequencing on 207 samples collected from symptomatic *P. falciparum* malaria cases along the Ecuador-Colombia Pacific Coast between 2013 and 2017 (Fig 1; S1 Table). The Colombian data set includes 151 samples obtained from venous blood

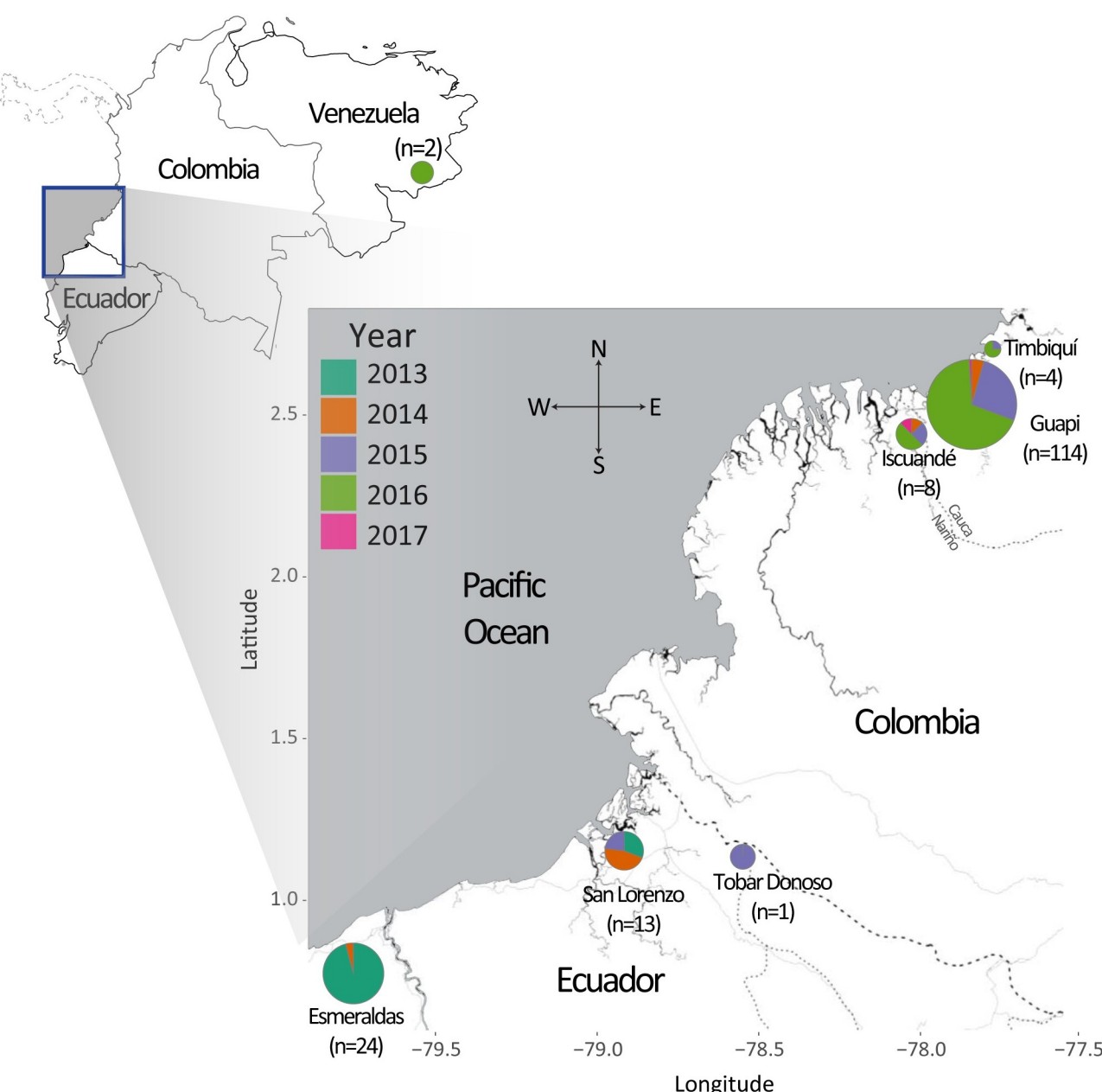

**Fig 1. Geographic and temporal distribution of 166 monoclonal *P. falciparum* samples collected along the Pacific Coast of Colombia and Ecuador from 2013 to 2017.** Samples in Colombia originated in three municipalities (Santa Bárbara de Iscuandé in Nariño and Guapi and Timbiquí in Cauca) and were diagnosed at the Guapi diagnostic microscopy post, with the exception of two monoclonal samples that originated in Venezuela. Pie chart divisions are colored by collection year and the area of the pie chart is proportional to the per-location sample count. Samples in Ecuador originated in three sites: Esmeraldas, San Lorenzo and Tobar Donoso. Esmeraldas and San Lorenzo are located 120 kms apart by road and Tobar Donoso is located 40 km east of San Lorenzo but there is no road to reach that locality. The number of samples indicates the number of high-coverage, monoclonal samples, which were used in all subsequent analyses. Map tiles by Stamen Design Toner style, under CC BY 3.0. Data by OpenStreetMap, under OdbL: http://maps.stamen.com/toner/#9/2.6015/-77.5003.

(leucocyte-depleted) from a 2014–2017 study that drew from the Guapi diagnostic microscopy post, which serves a range of interconnected rural and urban communities along the river networks of Guapi and neighboring municipalities [19]. When possible, travel history was documented, and from these data, three sampled infections were assumed to have originated in

Venezuela, with infected individuals traveling to the Guapi region for gold mining. In total, Colombia registered 104,074 *P. falciparum* cases between 2015–2017, from which approximately 92,000 were diagnosed in the four departments of the Pacific Coast region [26–29], and 3,920 in the municipalities of Guapi and Timbiquí (Cauca Department) and Santa Bárbara de Iscuandé (Nariño).

The Ecuador data set contains 56 samples collected at two study sites between 2013–2015: Esmeraldas and San Lorenzo. During these three years, Ecuador registered 396 cases of *P. falciparum* malaria, and 88% of the infections originated from these two Pacific Coast sites [30]. DNA from these samples was extracted from whole venous blood or filter paper blood spots and underwent selective whole-genome amplification prior to sequencing [31].

We restricted our downstream analysis to samples that had variant calls with greater than or equal to 5X coverage for at least 30% of the genome. Due to the different extraction and sequencing methods, the success rate for the two data sets varied, leaving 139, 38, and 3 samples from Colombia, Ecuador, and Venezuela, respectively. From within these high-coverage samples, we identified a high fraction of putatively monoclonal samples (126, 38, and 2 samples originating from Colombia, Ecuador, and Venezuela, respectively). This final set of 166 high quality, monoclonal samples was retained for further analysis.

## The Pacific Coast parasite population shows high relatedness and a large proportion of clonal relationships

A previous analysis of 12 polymorphic microsatellites and 272 SNPs found two genetically distinct populations of *P. falciparum* in South America, separated by the Andes mountain range [16]. To explore this pattern at a whole-genome scale, we conducted a principal component analysis (PCA) that combined our Pacific Coast samples with whole-genome sequenced samples from Guyana in the eastern portion of the continent [13]. The results support the previously observed separation of eastern (Guyana) and western (Colombia-Ecuador) parasite populations (S1 Fig). This reinforces the evidence for strong structure at the continental scale in the South American *P. falciparum* population, underlining the potential for sub-regional elimination planning.

We next explored structure within the Western population, where prior studies using microsatellites and SNP panels have documented connectivity between Colombia, Ecuador, and potentially Peru, including the maintenance of clonal lineages along the Colombian coast for at least eight years [17,18,32]. In concordance with these studies, our PCA found no meaningful separation between Colombia and Ecuador samples (S1 Fig). To further dissect population structure across the region, we analyzed relatedness between parasite pairs by calculating identity by descent (IBD). Among all pairs of parasites originating from this region, fractional IBD is markedly higher than the relatedness values observed in other global populations[33], even other low transmission areas like Guyana and the Greater Mekong Subregion (Fig 2A) [13,34]. The median IBD between samples is 0.29, which is on par with the expected relatedness between half-siblings in a fully outbred population (0.25). Median IBD within Colombia alone is similar to this region-wide estimate (median IBD = 0.27), whereas median IBD in Ecuador is extremely high (median IBD = 0.76). It is worth noting that the Ecuador data set was enriched for samples collected during an outbreak in Esmeraldas that was dominated by a single clone [35] (E3; Fig 2B). This inflates the IBD estimate and highlights how robust genomic inference needs to account for biased sampling. IBD between Colombia and Ecuador is also high (median IBD = 0.36), supporting the hypothesis of high population connectivity throughout the Pacific Coast Region.

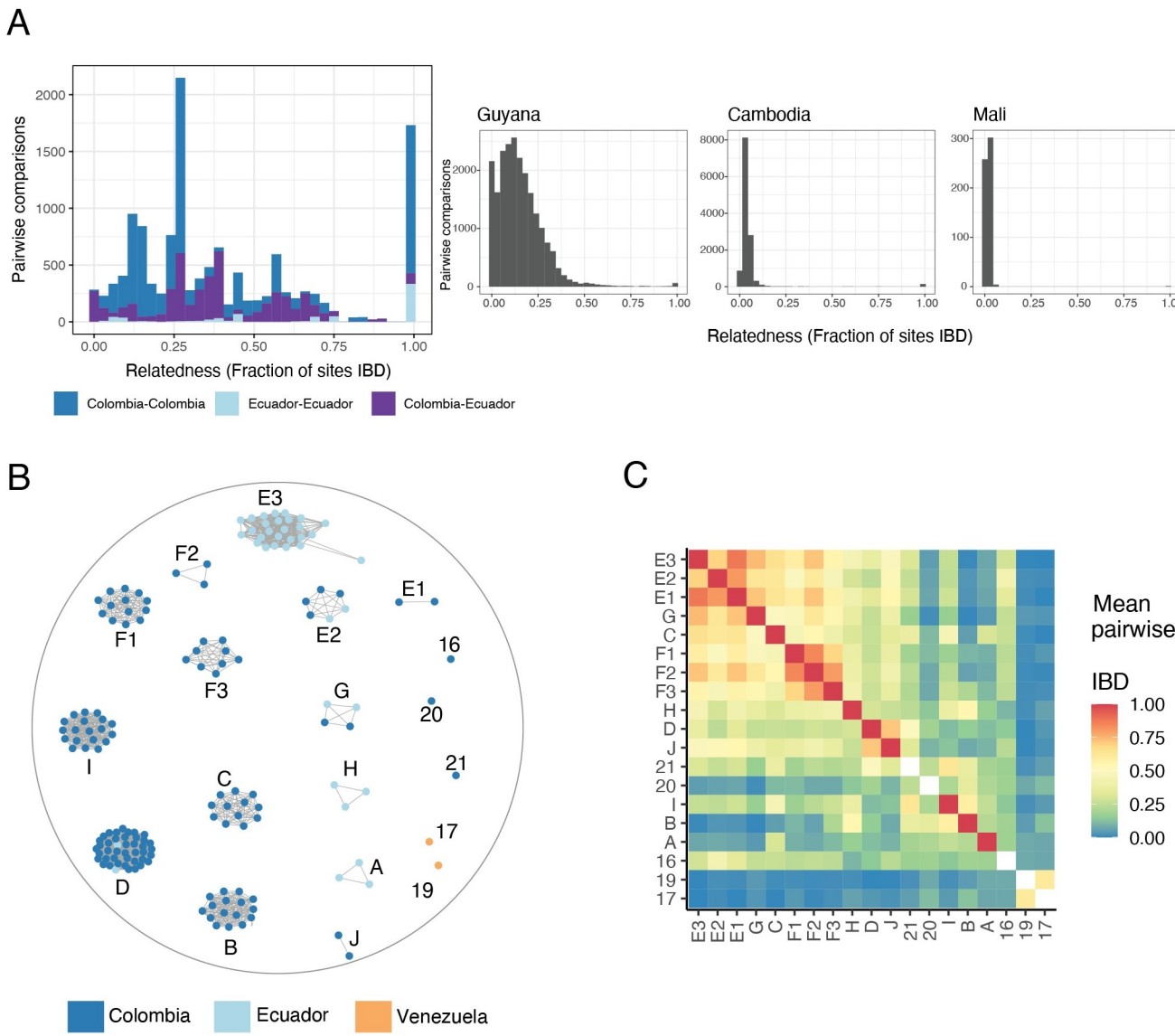

**Fig 2. Relatedness between parasites along the Pacific Coast of Colombia and Ecuador.** (A) Distribution of pairwise IBD in different transmission intensities: Pacific Coast of Colombia and Ecuador (2013–2017), Guyana [13], Cambodia [33] and Mali [33]. Samples from Western Cambodia belonging to the clonal expansion of K13 C580Y-harboring samples were excluded. (B) Network of 166 monoclonal samples collected from Colombia or Ecuador (2013–2017). Edges connecting nodes correspond to IBD≥0.99. (C) The mean IBD between clusters ranges from 0 to 0.89 and shows the presence of two larger "superclusters" at IBD>0.8. Two Venezuelan samples (17 and 19), show low IBD with Colombian and Ecuadorian samples but intermediate IBD with each other. Single samples (numeric labels) are missing diagonal values as no intra-cluster comparisons could be made.

In addition to this overall high level of relatedness, we observed 1,737 parasite pairs that had a clonal relationship (IBD > = 0.99). We grouped clonal samples into clusters, members of which can be separated by *de novo* mutations but not by recombination events (Fig 2B). In total, we identified 19 distinct genomic lineages. Of these, 14 were clusters containing two or more samples, while five—including the two Venezuelan samples—are singletons observed only once in the data set. The 14 clonal clusters range in size from two to 43 samples (S2 Table). Eight contain only samples from Colombia, three contain only samples from Ecuador, and three contain samples from both countries (clusters D, E2 and G). Relatedness between

clonal clusters is variable (Fig 2C), which explains the spikiness apparent in the pairwise IBD distribution (Fig 2A). For example, members of cluster B and cluster D are IBD across 14% of the genome. These clusters are both large, leading to 688 pairwise comparisons with IBD of approximately 0.14.

To further understand relatedness at a sub-clonal scale, we iterated the clustering algorithm across a range of IBD thresholds (0.1–0.9; S2 Fig). This identified two highly-related "superclusters" containing clonal clusters that connect at IBD ≥ 0.8. For ease of identification, we named multi-member clonal clusters with letters and singleton genomes with numbers. We denote the clusters forming the two superclusters as E1-E2-E3 and F1-F2-F3 (Fig 2B).

## IBD analysis resolves the population structure of the Pacific Coast Region

With the exception of Taylor et al. (2020) [32], previous analyses of *P. falciparum* population structure along the Pacific Coast have relied not on IBD but on alternative approaches such as admixture analysis and neighbor-joining trees. Of these, IBD most directly captures the relationships among samples by directly measuring relatedness. To evaluate the value added by whole-genome IBD analysis, we compared our IBD results to those obtained with other methods.

First, we used the tool ADMIXTURE [36] to delineate related groups within the data set. The highly inbred and largely clonal structure of the Pacific Coast population does not arise from admixture in the population genetic sense, however, the genomic clustering algorithms employed in tools like ADMIXTURE and STRUCTURE are capable of detecting potentially relevant groupings within the data. Knudson et al (2020) [19] recently took such an approach and used STRUCTURE with 101 genotyped SNPs from the same 2014–2017 Colombia samples. Here, we applied the related tool ADMIXTURE to our WGS data and employed best practices, without informing the model with our IBD results. This approach assigned the Colombia samples to five qualitatively distinct groups that correspond with the four largest clonal clusters (B, C, D and I) and one supercluster (F1-F2-F3) in the IBD analysis (Fig 3A). The ADMIXTURE groups recapitulate the three STRUCTURE groups identified in Knudson et al (2020) [19] (S3 Fig) and resolve additional sub-structure, possibly as a result of using whole-genome data. The ADMIXTURE classification, however, is affected by cluster frequency, which might not reflect true evolutionary or demographic patterns. To demonstrate this effect, we repeated the analysis after slightly altering cluster frequency on the order of what we expect to see with small localized outbreaks or uneven sampling. One cluster (C) was decreased from a frequency of 0.09 to 0.016 and a second (E1) was increased from a frequency of 0.015 to 0.06. This amount of simulated drift was sufficient to change which clonal clusters are described as "pure" versus "admixed" (S4 Fig). As compared to ADMIXTURE and STRUCTURE analysis, IBD clustering therefore provides superior interpretability, resolution, and consistency.

We next analyzed the data using genetic distance (or the complement of identity by state; 1-IBS) methods and visualized these data with the common approach of creating a neighbor-joining (NJ) tree (Fig 3B). The NJ tree captures the identity of the clonal clusters, but it cannot accurately represent the relationships between clusters because they arose through recombination of standing variation, not divergence from a common ancestor. For instance, cluster E3 is more highly related to cluster F2 (mean IBD = 0.72) than to cluster F1 (mean IBD = 0.59) or cluster F3 (mean IBD = 0.56). While these relationships are apparent on the IBD heatmap (Fig 2C), the branch lengths on the NJ tree appear comparable for all three comparisons. The discrepancy is caused by the visualization approach, not differences in IBD vs IBS calculations, as these two measurements are highly correlated at the cluster level (Pearson's $r$ = -0.96; S5 Fig).

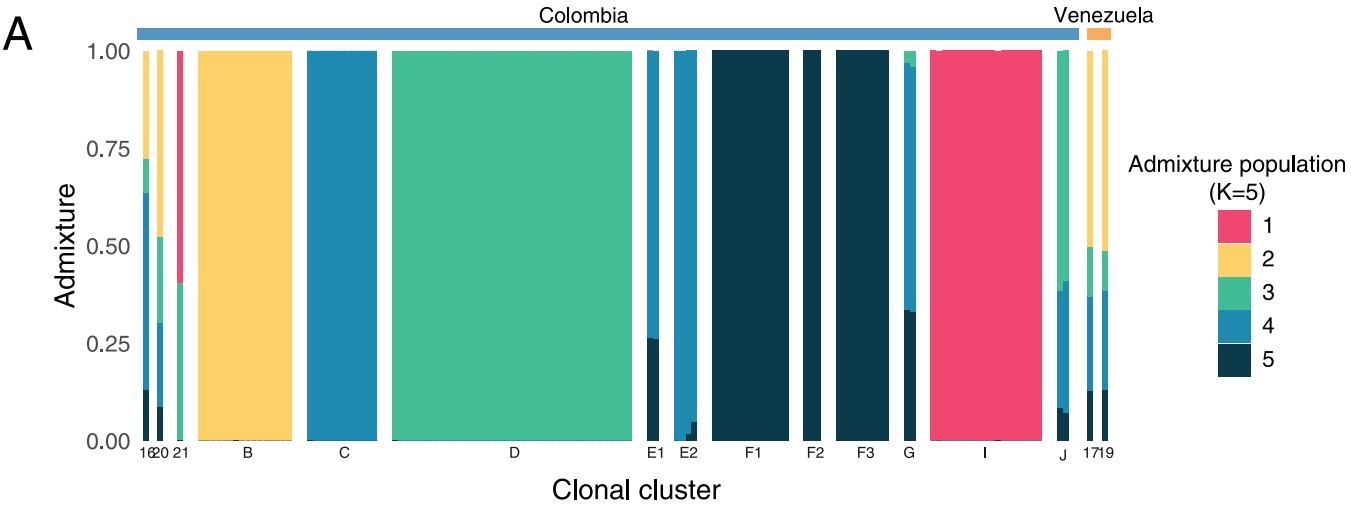

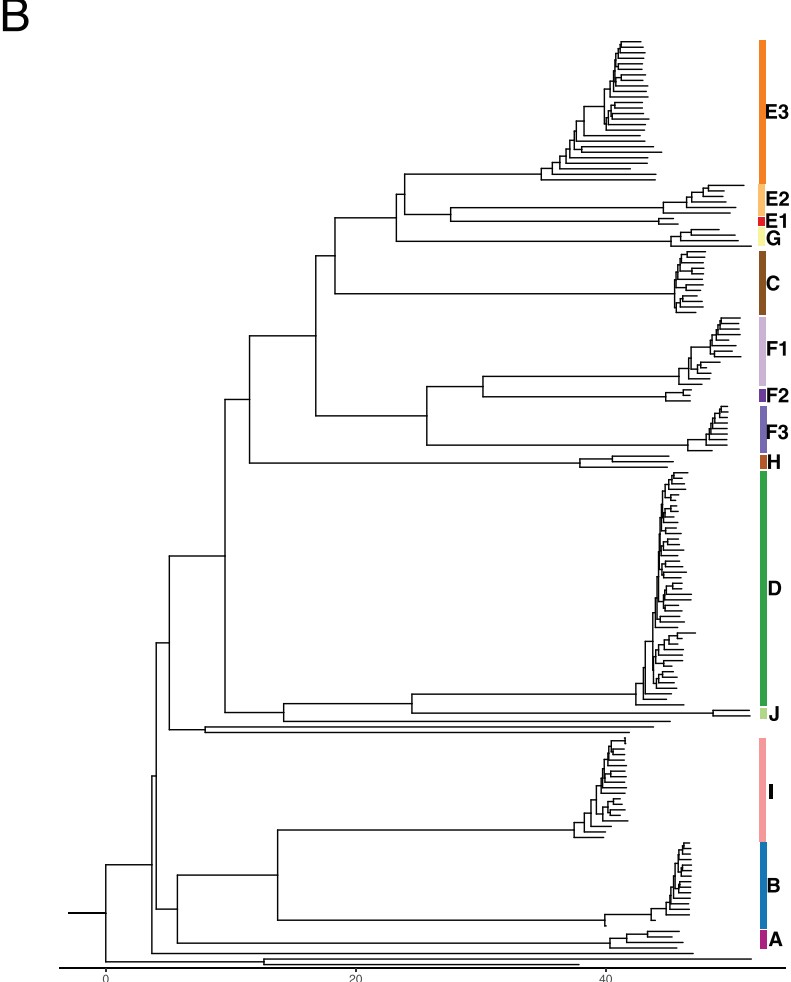

**Fig 3. Comparison of an IBD approach to genetic distance analysis and visualization methods.** (A) ADMIXTURE analysis of the 2014–2017 Colombia-Venezuela samples identifies five major groups, which correspond with the four largest clonal clusters (B, C, D, and I) and one highly related supercluster (F1-F2-F3) in the IBD analysis. In addition to being less granular, these results are dependent on cluster frequency (S4 Fig) and their interpretation is less straightforward than relatedness analysis. (B) Depicting cluster relationships with a neighbor-joining tree based on genetic distance shows more fine-scaled inter-cluster relationships than ADMIXTURE, however, these are qualitative whereas the IBD heat map in Fig 2C provides quantitative relatedness estimates.

It is worth noting, however, that only IBD—and not IBS—enables direct comparisons across studies and populations [37]. A heatmap for displaying IBD relationships may not be tractable with larger sample sizes and more outbred populations. In these instances, dimension reduction approaches like PCoA would be practical. Watson et al. (2020) [38] have recently laid out considerations for rigorously applying these methods to *Plasmodium* data sets.

## Six clonal lineages have persisted for a decade or longer within the Pacific Coast Region

We next explored the wider spatial and temporal genetic structure of the region by combining our 166 WGS samples with 325 monoclonal Colombian parasites longitudinally sampled between 1993 and 2007 that were previously genotyped at 250 SNPs [17,32]. To assess whether an IBD approach could be applied robustly to these distinct data sets, we repeated our IBD and clustering analysis with a modified protocol that incorporated confidence intervals to accommodate the high proportion of missing calls at the 250 SNPs in some whole-genome sequenced samples (20% of WGS samples were missing calls at ≥50% of the 250 sites).

Overall, we found that the panel of 250 SNPs provided comparable results to those obtained with genome-wide data. The analysis based on 250 SNPs initially identified 16 distinct genetic (clonal) backgrounds, which included 12 with multiple members and four singletons. Eleven of the 12 multi-member clonal clusters corresponded uniquely to a single WGS clonal cluster, and the remaining one encompassed three highly related WGS clusters (E1, E3, and G). The mean IBD point estimates between samples within this latter cluster was lower than that for the other 15 clusters (0.91 versus >0.99), and upon visual inspection, there was strong support for breaking this final group into the same three smaller clusters identified with WGS data (S6A Fig). With the sparser SNP data, the approach had lower power than a genome-wide analysis. Nevertheless, the two data sets provided IBD point estimates between clonal clusters that were highly correlated (Pearson's r = 0.93; S6B Fig).

We found that six of the multi-member clonal clusters sampled in 2013–2017 held clonal relationships with parasites sampled prior to 2008 (Figs 4 and S6C). The longest identified lineage (cluster E3) persisted back to at least 1999 and was the major driver of an outbreak in

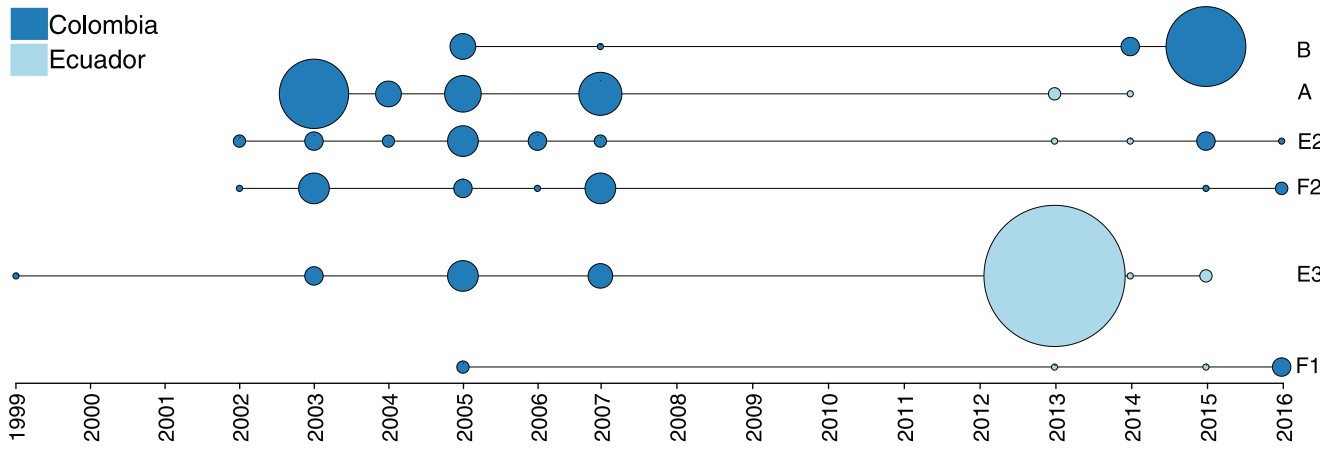

**Fig 4. Spatiotemporal distribution of long-persisting clonal clusters across Ecuador and Colombia.** A combined analysis of recent whole-genome sequencing data (2013–2017) and older longitudinally sampled parasites genotyped at 250 SNPs (1993–2007; N = 325 [32]) identified six clonal clusters that have persisted in the region for a decade or longer. Four of these clusters were sampled in both Colombia and Ecuador, albeit not within the same year. Clonal clusters detected in a given year are depicted by the vertices with the size of the vertex corresponding to the number of samples. S6C Fig breaks down the clonal components by within-country location.

Ecuador in 2013 [18]. Four of these persistent lineages incorporated samples from both Colombia and Ecuador.

## Known mutations conferring drug resistance are common throughout the region

The continued sampling of clonal lineages provides further evidence that the Pacific Coast *P. falciparum* population has been characterized by a low population-level recombination rate for at least the last two decades. Evolutionary theory predicts that selection will not be highly efficacious under these conditions, so we were therefore interested in exploring how recent selection has progressed in the region. Multi-drug resistance is documented throughout the area, but antimalarial use—and therefore specific selection pressures—historically differed in Colombia and Ecuador (Fig 5A). Prior to the mid-1950s, both countries relied primarily on chloroquine. After emergence of clinical resistance in the 1950s, Colombia began replacing chloroquine with pyrimethamine, first as a monotherapy and, later in the 1980s, in combination with either sulfadoxine or sulfadoxine and chloroquine (SP, SP+chloroquine) [39,40]. Resistance to SP arose as early as the 1980s in eastern Colombia, and use of the drug throughout Colombia was greatly reduced after 2006. Contrary to this, Ecuador continuously used chloroquine as the frontline malaria treatment until its replacement with artemisinin-combination therapies in the mid-2000s [3].

Given these specific drug regimes, we first assessed the prevalence of known resistance haplotypes in four genes that have strong prior evidence of contributing to drug resistance: Chloroquine-resistance transporter (*crt*, Pf3D7_0709000), dihydrofolate reductase (*dhfr*, PF3D7_0417200), dihydropteroate synthetase (*dhps*, PF3D7_0810800), and multidrug resistance protein 1 (*mdr1*, PF3D7_0523000). Mutations in a fifth gene, *kelch13* (PF3D7_1343700), have been associated with resistance to the antimalarial artemisinin, however, in our data set no known resistance mutations were present in this gene, so it was excluded from further analysis.

Using all samples, we calculated the frequencies of these known resistance haplotypes at the country level. Overall, there was a high proportion of resistance-associated mutations in both Colombia and Ecuador, although wildtype (drug sensitive) haplotypes were present at appreciable frequencies for both *dhfr* and *dhps*. No sample contained wildtype haplotypes at *crt* or *mdr1*, and no sample contained the CRT C350R mutation, which has been documented as restoring chloroquine sensitivity in French Guiana [41]. Both samples originating from Venezuela contained highly resistant haplotypes at all successfully genotyped loci, and were the only samples containing the DHFR triple mutation, highlighting important differences between the two regions (Fig 5B). The different patterns in Colombia and Ecuador may reflect the different historical drug usage, but they are also affected by the highly clonal structure of the population, which leads to repeated—and likely stochastic—sampling of the same genomic backgrounds. It is therefore unlikely that these raw allele counts accurately reflect selection. For instance, a single clonal cluster (E3) dominates the Ecuador sample set because of its high prevalence during the 2013 Esmeraldas outbreak (Figs 2 and 4). However, it is unknown whether this clone reached high frequency due to high intrinsic fitness or because it was simply present when ecological conditions became conducive for an outbreak.

In addition to calculating population-level allele frequencies, we mapped the resistance mutations to our clonal clusters (S7 Fig). This mapping suggests that frequent *de novo* mutation is not a major force structuring drug resistance at these loci as all samples within a clonal cluster contained matching haplotypes. Further, a subset of 17 pre-2007 samples from four of the six persistent clonal lineages (Fig 4) were previously genotyped at *crt* and *dhps*, and all genotypes were concordant through time.

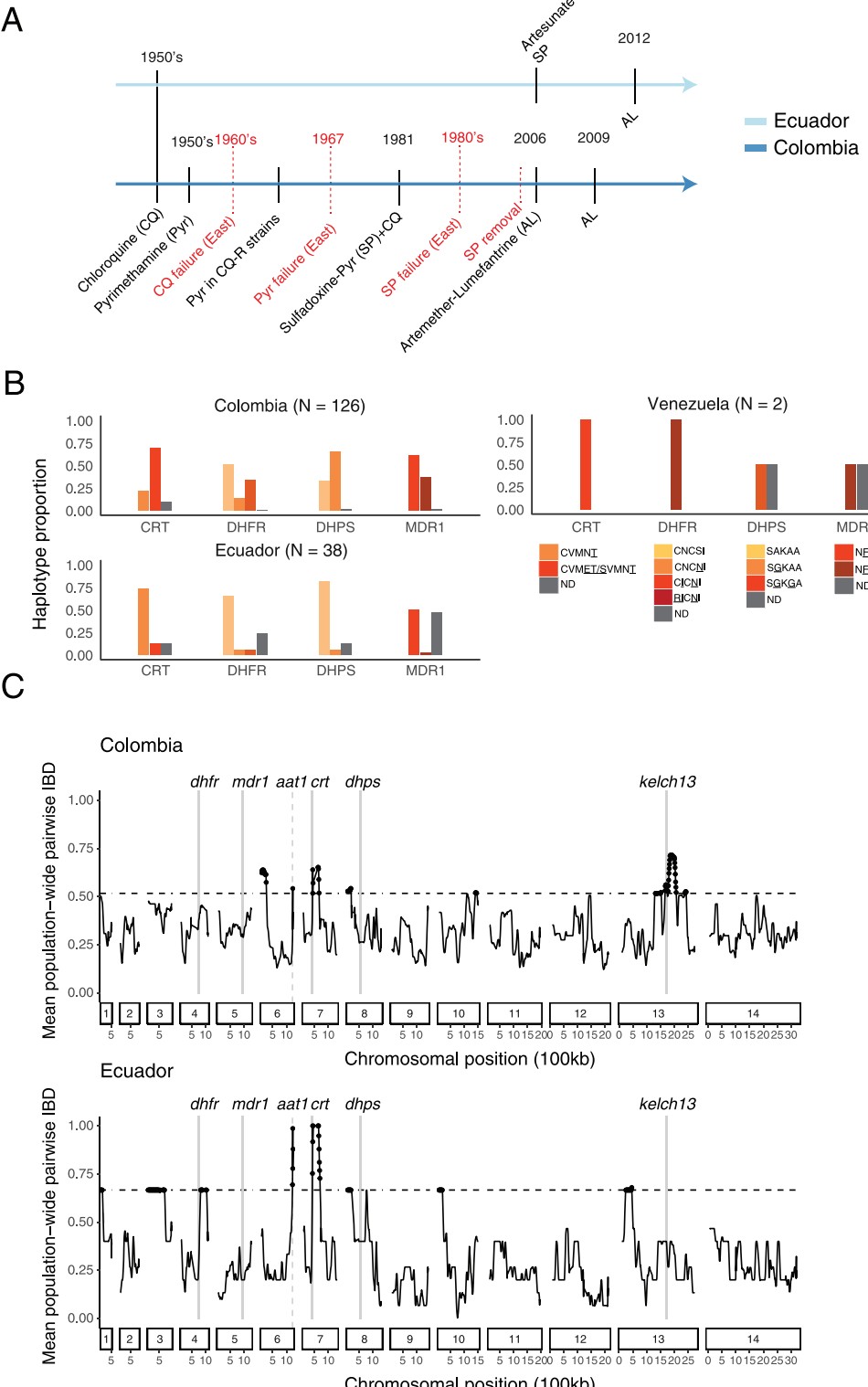

**Fig 5. Drug selection across the Pacific Coast Region.** (A) Timeline of official antimalarial usage for treatment of *P. falciparum* malaria (black) and documented emergence of antimalarial resistance (red) in Ecuador (light blue) and Colombia (dark blue). Data are based on epidemiological information from the respective country-level Institutes of Health. (B) Drug-resistance haplotypes in the 166 monoclonal samples collected 2013–2017, colored by wild-type (WT) allele, or by the drug resistance haplotype shown as the amino acid sequence, with an underline referring to the known

functional mutations, or in gray for haplotypes not-determined from read coverage (ND). Drug-resistance haplotypes at the cluster level are available in S2 Table. (C) Mean pairwise IBD within 50-kb overlapping windows across the genome (y-axis) for Ecuador and Colombia. IBD calculations were made using one representative sample per clonal cluster. Horizontal lines mark the 95th percentile for Ecuador (bottom) and Colombia (top). Vertical gray lines mark the chromosomal positions of genes with known involvement in antimalarial resistance.

## Shared IBD segments identify putative selective sweeps at known and novel targets

We next examined shared ancestry both genome-wide and around these specific resistance loci to understand how mutation and recombination have shaped patterns of selection. To investigate intrachromosomal patterns of shared ancestry, we calculated the mean pairwise IBD within overlapping 50-kb windows. Unlike the allele frequency estimates, distinct genomic backgrounds were included only once in the calculations to minimize the impact of stochastic re-sampling. While migration between countries is observed, we initially analyzed the two data sets separately to look for potential regional differences. As a means of focusing on genomic regions that likely experienced the strongest selective sweeps, we examined windows with mean relatedness above the 95th percentile. Six and nine peaks crossed this threshold in Colombia and Ecuador, respectively, with three of these regions shared by both countries (Fig 5C). Two of these shared peaks contain genes implicated in resistance to chloroquine, a drug that is known to have imposed strong selection on *P. falciparum* in the 20th century. While sample sizes are small and the effect of stochastic sampling is likely strong, it is interesting to note that the mean IBD within both windows is higher for Ecuador, where chloroquine selection took place for a longer period of time (Fig 5A).

One chloroquine-associated IBD peak begins at *crt* on chromosome 7, where a strong selective sweep is known to have occurred in South America leading to the fixation of the large-effect variant K76T along the Pacific Coast [11,12,19,35,42]. As we discuss further below, we observe an extreme depletion of segregating polymorphisms around *crt*, making the locus of selection difficult to pinpoint. The observation that *crt* is on the edge rather than the center of the IBD peak could be an artifact stemming from this low resolution or it could reflect an additional sweep that occurred after the K76T fixation.

A second chloroquine-associated peak found on chromosome 6 is a novel result in this geographic region. The peak contains (in the case of Ecuador) or is adjacent to (for Colombia) amino acid transporter 1 (*aat1*, Pf3D7_0629500), which has been identified as mediating resistance to chloroquine and other drugs [43,44] and as being under selection in natural populations [45,46]. In this data set, we observe a high-frequency derived allele that causes a nonsynonymous serine to leucine change at amino acid 258 (S258L), which falls within the protein's predicted transmembrane domain. Only one Pacific Coast sample (SPT26239) carries the ancestral allele. This sample is an outlier in our data and shows the lowest overall relatedness to the other Pacific Coast samples in our collection. Both Venezuelan samples also exhibit the ancestral allele, evidence that this putative selection signal at *aat1* may be population-specific rather than continent-wide like *crt* (S8 Fig). To further understand the timing of selection, we examined the frequency of this mutation in 16 Colombian samples collected prior to 2005 that were whole-genome sequenced together with the sample set used for the longitudinal analysis of clonal persistence. The older samples were fixed for the known resistance-conferring K76T CRT mutation, and the ancestral AAT1 S258 allele was present in 25% of these early samples. In sum, while the genomic IBD scan cannot definitively pinpoint the causal allele driving selection, these results provide the first evidence of potential *aat1*-mediated chloroquine resistance in South America.

The third IBD peak within the 95th percentile for both countries is located on chromosome 8. This peak does not contain any known selection or drug targets, but it does include high-frequency coding variants in a gene associated with the proteasome/ubiquitination pathway (SENP2; PF3D7_0801700), which was previously identified in a study of artemisinin-driven selection in Southeast Asia [47].

Joint analysis of Colombia and Ecuador WGS data yielded similar results to the country-specific analyses. There are eight peaks above the 95th percentile (S9 Fig), and all but one overlap outlier regions in at least one country-level analysis. In addition to the peaks discussed above, two other large peaks stand out in both the joint and Colombia-specific analyses: one is on chromosome 13 and the second covers an additional region of chromosome 6. The chromosome 13 peak contains *kelch13*, a gene for which several amino acid variants are associated with resistance to the antimalarial drug artemisinin. In our data set, however, the only identified Kelch13 variant is a common, wide-spread polymorphism (K189T) that has shown no evidence of carrying a phenotypic effect [48]. We therefore hypothesize that the driver of the sweep is an alternative locus. Candidates include genes with prior evidence of drug-associated phenotypes such as tyrosine kinase-like protein 3 (PF3D7_1349300) [49] and lysine—tRNA ligase *(krs1*; PF3D7_1350100) [50–52]. In this data set, both of these genes contain high-frequency variants that alter the coded protein sequence. In addition, a key gene involved in *Plasmodium* adaptation to novel vectors (*pfs47*; PF3D7_1346700) falls in the center of this window [53,54]. While we did not identify any high-frequency coding segregating variants in this gene, the pattern could reflect the past fixation of selected alleles or incomplete selection on expression-level variation mediated via upstream or downstream polymorphisms, which we did not analyze with this short-read data.

In contrast to *crt*, the genomic regions around the three other focal drug resistance genes (*dhfr*, *mdr1*, and *dhps*) do not show IBD patterns indicative of universal hard selective sweeps (Figs 5C and S9). A genomic segment bordering *dhfr* shows elevated IBD in Ecuador, suggesting strong selection, but the same signal is not present in Colombia or the combined data set. The high frequencies of known resistance-conferring alleles within these genes therefore suggest that selection may have mainly occurred through soft or incomplete sweeps. To evaluate this hypothesis, we examined the diversity of haplotype sequences in the 30-kb regions surrounding these loci (Fig 6). We selected one representative sample from each clonal cluster and additionally included the 16 pre-2005 Colombian samples mentioned above. We did not attempt to statistically evaluate these regions with haplotype-based selection tests due to the limited number of distinct genomic backgrounds and the presence of both geographic and temporal structure in the dataset.

Haplotypic diversity around *crt* is very low, consistent with there having been a hard selective sweep at this locus (Fig 6). We observe only three distinct 30-kb haplotypes in the expanded data set of 33 genomes. One haplotype contains two *crt* mutations (CVM<u>ET</u>). This double-mutant haplotype was the only one found in the 16 pre-2005 Colombian samples showing that selection had already occurred before this time. A large proportion of the post-2013 samples from Colombia and Ecuador instead carry one of two 30-kb haplotypes that carry a single *crt* mutation (CVMN<u>T</u>; Fig 5B). These single-mutant haplotypes differ by only one nucleotide change, making it probable that they represent the same ancestral haplotype separated by a single, recent mutation event (Fig 6). The double mutant did not occur on the single-mutant background, showing different origins and separate selection events for these two haplotypes [11,12,42,55].

Contrasting with the haplotypic pattern around *crt*, combinations of resistance mutations in *mdr1*, *dhfr*, and *dhps* appear on multiple haplotypic backgrounds in both the pre-2005 and post-2013 data sets (Fig 6). This could arise from either multiple *de novo* mutation events, or

### Chloroquine resistance transporter (*crt*)
### Pf3D7_0709000

### Multidrug resistance protein 1 (*mdr1*)
### Pf3D7_0523000

### Dihydrofolate reductase-thymidylate synthase (*dhfr*)
### Pf3D7_0417200

### Pyrophosphokinase-dihydropteroate synthase (*dhps*)
### Pf3D7_0810800

**Fig 6. Haplotypes surrounding known drug resistance mutations in *crt*, *mdr1*, *dhfr*, and *dhps* show varying degrees of diversity.** Plots display segregating SNPs within the 30-kb flanking regions around mutations of interest within four genes involved in antimalarial drug resistance. One high-coverage sample from each clonal cluster is displayed (rows). SNP positions were removed if they were within five nucleotides of a called indel or if they displayed a high rate of heterozygous calls within monoclonal samples (Materials and Methods). Calls are colored based on matching (light grey) or mismatching (green) the 3D7 reference. Calls are marked as missing (dark grey) if they were

heterozygous or had fewer than five reads. Black outlines at alleles coding for *dhps* position 437 indicate that 3D7 contains the A437G mutation; wildtype calls are therefore green at this position. Nonsynonymous mutations of interest are depicted with a solid vertical red line. One synonymous mutation at codon 540 in *dhps* is depicted with a dashed vertical red line. Represented genes and downstream codon changes are (top to bottom): *crt* (72 and 75); *mdr1* (1042 and 1246); *dhfr* (50, 51 and 108); and *dhps* (437 and a synonymous mutation coding for 540). S2 Table lists the mutations detected in the clonal clusters.

weaker selection (relative to that at *crt*), which allowed time for mutations to recombine onto different backgrounds. Of the four loci, only *dhps* maintained numerous distinct wildtype haplotypes, suggesting that selection has been weakest at this resistance locus, consistent with the history of drug therapy in both countries.

Interestingly, at *dhfr*, wildtype alleles are common and found on 11 of the 17 Pacific Coast genomic lineages. All of these genomes, however, are identical across the surrounding 30-kb region. In contrast, variation is much higher between haplotypes carrying the single S108N mutation. Laboratory experiments have shown that the DHFR S108N mutation carries a fitness cost in the absence of pyrimethamine [56]. The loss of haplotypic diversity around the ancestral wildtype allele is consistent with there having been strong selection against this allele when the drug pyrimethamine was in use, followed by a re-expansion of the wildtype allele as pyrimethamine dosage was reduced in 2006 in Colombia and in 2011 in Ecuador, where SP was used intermittently as a second-line treatment [57]. This expansion of the wildtype allele parallels field observations of an increase in wild-type *crt* alleles [58,59] or the rise of compensatory mutations [41] following chloroquine cessation in other global populations.

In most instances, the two samples of Venezuelan origin carry haplotypes with distinct polymorphisms and even unique combinations of resistance alleles relative to Pacific Coast parasites. Although the sample size is too small to be conclusive, this pattern suggests that the Venezuelan *P. falciparum* population has experienced selection on independent *de novo* mutations. The one exception is *dhps*. At this locus, one Venezuelan sample carries a haplotype that matches a subset of Pacific Coast samples.

## Discussion

The epidemiological, ecological, and evolutionary dynamics of malaria are expected to change as disease transmission declines in response to control efforts, and these transitions may necessitate different analytical tools. Studying these dynamics in current areas of low transmission can therefore inform best practices for future genomic epidemiological studies and prepare us to track malaria decline more globally. Here, we used IBD metrics to gain insight into both the population structure and recent evolution of *P. falciparum* parasites along the Pacific Coast Region of Ecuador and Colombia, a region of low transmission. The results demonstrate that IBD accurately characterizes the highly clonal structure of this population, and may be useful for studying other parasite populations as control efforts advance. Other common analysis approaches such as PCA, ADMIXTURE, and Neighbor-Joining trees only partially recapitulate these results, and do not lend themselves to a straightforward biological interpretation in the same manner as measurements of relatedness between parasites.

Because relatedness estimates can be compared between populations and time points [37], an IBD approach further enables a more detailed understanding of global and temporal variation in inbreeding and clonality—even between areas like the Pacific Coast Region and Greater Mekong Subregion, which are often both included under the same "low transmission" umbrella. These comparisons have practical importance as several transmission-related phenomena can impact relatedness in a population. These include: local sub-structure, which can inform the spatial implementation of control strategies; the proportion of imported versus

endogenous parasites; and general transmission declines, for which genomic surveillance may prove a cost-effective means of assessment.

Our study incorporated new whole-genome sequence data from 166 monoclonal infections. This level of genomic resolution deepened our understanding of how selection has progressed in the region by enabling whole-genome scans and genetic analysis of large haplotype blocks. These two approaches provide strong support for there being at least two loci that experienced hard selective sweeps as a consequence of chloroquine pressure: *crt* and *aat1*. The phenotypic effects of *crt* mutations have been previously studied in South America, but *aat1* was only recently implicated in chloroquine resistance. This is the first evidence of this gene's potential role in the Americas and raises concerns regarding cross-resistance to other quinoline-based combination therapies. In contrast to *crt* and *aat1*, three other genes with known drug-resistance phenotypes show evidence of soft, rather than hard, sweeps. Given the small number of distinct genomic backgrounds and the long-term persistence of these haplotypes—most of which trace back a decade or longer—we cannot definitively differentiate between recurrent *de novo* mutations and recombination. Regardless of source, however, the persistence of multiple mutation-bearing haplotypes at *mdr1*, *dhfr*, and *dhps* versus the hard sweeps at *crt* and *aat1* highlights differences in the strength of selection connecting these drug-gene pairs. Interestingly, we also observe that the removal of drug pressure may have aided the expansion of wild-type *dhfr* haplotypes.

Our IBD analysis detected selection events (for instance, to chloroquine) that may be a half-century old and would have occurred under different transmission dynamics. However, the repeated sampling of parasites from persistent clonal lineages since at least 1999 shows that population-level recombination has remained low over the last two decades. Despite this, we find evidence that selection remains effective. In recent years, a single *dhfr* haplotype containing wildtype alleles has expanded onto multiple genomic backgrounds within the Pacific Coast population. Given the reduction of pyrimethamine usage in Colombia in 2006, this is likely a recent selective event. This observation contributes to a growing body of literature that demonstrates how *P. falciparum*'s evolutionary potential is maintained even when effective population size is small and within-host competition is low [60,61]. *Plasmodium* experiences extreme fluctuations in cell count over the course of its life cycle. These expansions and contractions cause the site frequency spectrum to differ from Wright-Fisher expectations [62] and create higher levels of standing variation than some measures of effective population size would suggest [63]. Here, we find that *de novo* mutation alone does not govern adaptation in small populations. Recombination also plays a role, and when selection is strong, even low population-level recombination is sufficient to passage new beneficial alleles onto multiple genomic backgrounds and through the population. Outcrossing can only occur when multiple genotypes are present in a mosquito's blood meal (COI > 1), and so transmission dynamics likely play a large role in governing evolutionary potential across this region. In both Colombia and Ecuador, localized outbreaks periodically drive infection counts well above baseline levels (API>10) and may increase the likelihood of outcrossing. Analyzing the temporal and spatial distribution of these events, determining their contribution to outcrossing events [35], and assessing the role of epistasis in driving clonal dynamics are the next steps for describing evolutionary trajectories in populations dominated by monoclonal infections. This will increase our capacity to anticipate the course of adaptation to drugs—and other novel interventions—as other populations near elimination.

The whole-genome data generated here also serve as validation for results obtained with other genotyping approaches. We found that a mid-sized panel of 250 SNPs [17] recapitulates the clonal clusters identified with whole genome sequencing and provides reasonable IBD point estimates for partially related parasites (S6B Fig). However, uncertainty can overwhelm

estimates of IBD-based relatedness based on sparse marker data. Moreover, samples from different data sets are liable to miss data at many markers, as they do here. Consequently, confidence intervals are critical for both maximal sample retention and for quality control: with confidence intervals, we can estimate relatedness as tolerantly as possible (e.g. estimate relatedness for samples that share any data) and then use the confidence intervals to filter highly uncertain estimates; without confidence intervals, we must resort to an arbitrary SNP cut-off (e.g. only estimate relatedness for samples that share data on at least 100 SNPs) and then hope that the estimates are reliable. With this data set, employing confidence intervals permitted the analysis of samples with as few as 42 SNP calls. We caution, however, that this number will vary among data sets—and even among samples—as the informativeness of sites depends on factors like population structure, linkage disequilibrium, and allele frequency.

Recent efforts in the malaria field have borne several amplicon panels with wide geographic breadth that can quickly and affordably genotype hundreds or thousands of samples [23,64,65]. While whole-genome sequencing will remain the bedrock of selection analyses, these methodological advances in targeted sequencing will facilitate the use of IBD analysis with confidence intervals for describing local population structure (as we do here) [66,67], measuring connectivity between populations [68], identifying likely importation events [69], and tracking changes in transmission [70]. These advances in genomic epidemiology are enhancing established malaria surveillance toolkits and enabling responses tailored to individual country's needs [23,64,71,72].

## Methods

### Ethics statement

Sample collection in Colombia (2014–2017) was approved by the ethics committee (Evaluation Report 127–14 and 003-021-16) of the Medical School of the Universidad Nacional de Colombia [73–75]. Sample collection in Ecuador was approved by the Ethical Review Committee of Pontificia Universidad Católica del Ecuador (approvals #: CBE-016-2013 and 20-11-14-01). Formal written consent was obtained from all participants.

### Sample collection

In Colombia, sample collection at the Guapi Health Post took place between 2014 and 2017 from individuals reporting malaria symptoms, and in diagnostic posts in the Guapi municipality (El Carmelo), as well as in Santa Bárbara de Iscuandé (Chanzará) and Timbiquí (El Cuerval) as previously reported [19]. Upon arrival, participants were diagnosed with malaria via microscopy, and after obtaining written informed consent, 2-5mL of venous blood were collected into ethylenediaminetetracetic acid (EDTA) vacutainer tubes (BD Vacutainer). Sample origin was determined through a travel history survey. Cases were coded as local if a patient had remained at the site throughout the previous two weeks and imported if they had spent the majority of the previous two weeks at an alternate location. National and international health research standards were considered, and the project was presented and discussed with the local health authorities. The present work corresponded to minimal risk.

Colombian samples from between 1993 and 2007 were collected in municipalities from four departments on the Pacific Coast: Tadó and Quibdó in Chocó, Buenaventura in Valle, Guapi in Cauca, and Tumaco in Nariño. Informative samples (N = 325) reported in this study were genotyped at 250 SNPs from blood spots collected on filter papers as reported in Echeverry et al [17]. In addition, 16 samples were used for whole-genome sequencing. Briefly, 5 mL of venous blood was collected, followed by adaptation to *in vitro* culture [76]. Genomic DNA was extracted using the Purelink extraction kit (ThermoFisher Scientific Waltham, MA, USA).

In Ecuador, samples were collected between 2013 and 2015 by Ministry of Health personnel from individuals reporting malaria symptoms, and in diagnostic posts in the San Lorenzo and Esmeraldas municipalities as well as in the locality of Tobar Donoso (border with Colombia). Upon arrival, participants were diagnosed with malaria via microscopy. Written informed consent was provided by study participants and/or their legal guardians and 2-5mL of venous blood were collected into CPD vacutainer tubes (BD Vacutainer). Alternatively, 2–4 drops of blood were spotted on 3M filter papers. Sample origin was determined through a travel history survey. Cases were coded as local if a patient had remained at the site throughout the previous two weeks and imported if they had spent the majority of the previous two weeks at an alternate location.

## Whole-genome sequencing

Samples collected in Colombia, belonging to the studies undertaken in Knudson et al., and in Echeverry et al., were sequenced at the Wellcome Sanger Institute, as part of the MalariaGEN *Plasmodium falciparum* community project [33]. An Illumina HiSeqX platform was used to generate 200-bp paired-end reads. Samples collected in Ecuador underwent selective whole genome amplification at the Harvard School of Public Health [31] before library construction with Nextera XT library kits and sequencing at the Broad Institute on an Illumina HiSeqX.

We aligned reads to the *P. falciparum* 3D7 v.3 reference assembly and called variants following the best practices established by the Pf3K consortium (https://www.malariagen.net/projects/pf3k). In brief, raw reads were aligned with BWA-MEM [77] and duplicate reads were removed with Picard tools. SNPs and indels were called with GATK v3.5 HaplotypeCaller [78]. Base and variant recalibration (BQSR and VQSR) steps were performed using a set of Mendellian-validated SNPs. Downstream analysis was limited to variants found in the core region of the genome, as defined by Miles et al [79]. To estimate the number of genomes per sample, we used TheRealMcCoil [80] with a set of 1955 variants spaced at roughly 10kb intervals throughout the genome. We removed all polyclonal samples from further analysis. For the purportedly monoclonal samples used in the downstream analysis, we masked any site that was called as heterozygous in >10% of samples and masked any individual call supported by fewer than five reads. Unless otherwise noted, we also excluded from analysis any variant within 5 nucleotides of a GATK-identified indel.

In multiple samples from the 2014–2017 Colombia data set, we found evidence of exogenous PCR amplicon contamination around the *kelch13* gene, and so this gene was masked from downstream analysis. We visually inspected samples in this region to determine if there was any remaining evidence for *kelch13* variants and results were cross-checked with previous genotyping that had been performed on these samples. We detected only one valid polymorphism in the gene, which codes for the known, common variant K189T.

## Population structure in the Pacific Coast

To determine the population structure of samples obtained in Colombia (2014–2017) and ancestral composition of the sympatric populations circulating in Guapi, previously reported in Knudson et al. [19], we combined Principal Component Analysis (PCA), as well as a Bayesian model-based model of admixture (ADMIXTURE) [36]. For PCA analysis, a VCF containing only non-singleton SNPs within the core genome of a data set with a single sample for each cluster selected from the complete data (30% 5x coverage genome-wide) was filtered to remove all positions with missing data. We then used PLINK software to prune for linkage disequilibrium (LD) on windows of 100 SNPs, with 10 SNP sliding windows and a pairwise correlation (R^2) of 0.1 [81,82]. We used the filtered VCF containing a single representative sample per

IBD cluster to have balanced allele frequencies leading to higher accuracy in the analysis. This pruning resulted in >11,000 high-quality SNPs left for analysis by principal components (S1 Fig).

For population structure analysis using ADMIXTURE, we followed best practices for determination of the optimal number of ancestral populations (K) [36]. A VCF file containing the samples from the 2014–17 sample set from Colombia filtered to contain only biallelic SNPs was used and with heterozygous positions masked. Additional filtering was performed to include only variants in >80% of samples. As with the PCA, LD-pruning was performed on windows of 100 SNPs, with 10 SNP sliding windows and a pairwise correlation (R^2) of 0.1 [81,82]. SNPs that appeared to be in LD were removed and the pruned VCF was used to generate the pedigree-like file output from PLINK required as input for ADMIXTURE. We then performed for a range of K = 1 to 10 ancestral populations and the optimal number was determined with the elbow method (S3 Fig).

## IBD analysis

We analyzed fractional relatedness as identity by descent for putatively monogenomic WGS samples. To determine an appropriate coverage threshold, we investigated both the robustness of IBD calls and the appearance of analysis/data artifacts as coverage decreased (S10 Fig). We found a slight deflation in IBD estimates as samples approached 30% 5x coverage, but overall, performance was consistent across the 30%-97% coverage range. Below 30% 5x coverage, we began to observe reduced performance (eg, increased variance among repeated measures) and analysis artifacts (eg, spurious clique formation when performing igraph clustering). We therefore included only samples with > = 5x coverage for >30% of the genome in the IBD analysis. We independently arrived at a similar coverage cutoff with another recent WGS data set that employed IBD analysis [83]. We estimated IBD on whole-genome samples with a set of 16,460 SNPs using hmmIBD [84]. The SNPs included in the analysis satisfied the following requirements: (1) found within the core genome as defined by Miles et al, 2016 [79]; (2) >5nt from any GATK-called indel; (3) called in at least 80% of samples; (4) minor allele frequency > = 0.05; (5) called as heterozygous in <10% of declared monoclonal samples. Individual calls with <5x read support were marked as missing, and samples that had <25% of the genome with 5x read coverage were excluded from the analysis. We estimated population-level allele frequencies in three ways: (1) using the full dataset (default parameters), (2) using a sample set with only a single representative per clonal cluster; and (3) using only samples from the pre-2008 data set. For genome-wide IBD estimates, the differences among these methods were minimal, and unless otherwise stated in the text, the default estimates were used in downstream analyses. We performed clustering based on the fraction of sites called IBD as estimated with Viterbi algorithm (fract_vit_sites_IBD) by constructing an adjacency matrix in the R package igraph [85].

We performed a separate extended IBD analysis that included 325 additional pre-2008 samples from Echeverry et al (2013) that were genotyped at 250 SNPs using a GoldenGate panel. The combined analysis included the 250 GoldenGate calls made for the Echeverry samples and GATK calls made at the same positions for the whole-genome sequenced samples. GoldenGate calls were decoded using genotyping results from a 3D7 lab strain that were then compared to the PlasmoDB 3D7 reference sequence, as well as Dd2, Santa Lucía, HB3 and 7G8 [86,87]. Sixteen samples from Colombia had undergone both GoldenGate genotyping and whole genome sequencing, and we confirmed that the two platforms made identical calls in all cases with the exception of one site that was then masked from analysis. In addition to decoding, we reordered some SNPs whose names we discovered were previously missordered.

The extended analysis relied upon confidence intervals around IBD-based relatedness estimates, which were computed using the statistical framework described in Taylor et al. 2019 [37]. After removing one SNP for which there was no data among the WGS samples and seven WGS samples that had no data among the remaining 248 SNPs, we were left with 248 SNPs and 519 samples with a lot of missing data. To maximize sample retention, we estimated relatedness as tolerantly as possible (i.e. using all samples that share any data). To ensure quality control, we then used confidence intervals to filter uncertain estimates. As in Taylor et al. 2020 [32], confidence intervals were also used to circumvent an arbitrary SNP cut-off for clonal classification and the igraph package in R [85] was used to construct clonal components (referred to as clusters in the results). Since the igraph package does not support the construction of clonal components using samples with missing relatedness estimates, those samples were removed, leaving 496 samples with 122,760 relatedness estimates based on data on 12 to 248 SNPs. A sample pair was considered clonal if its 95% confidence interval included one and exceeded 0.75. This clonal definition, which is more stringent than that used before (95% confidence interval includes one; Taylor et al. 2020 [32]), was necessary to minimize the number of cliques within clonal components (ideally all clonal components should be cliques—fully connected subgraphs—but they are not due to greater uncertainty among relatedness estimates). The same principle (minimizing cliques within components) was used to break down a clonal component containing six cliques into three clusters: the largest containing two cliques, the other two continuing a single clique each (S6A Fig). For further details of the extended analysis see https://github.com/aimeertaylor/ColombianBarcode/blob/master/Code/Extended_analysis/Analysis_summary.Rmd.

As a final aside, we did a marker-reordered analysis of Taylor et al. 2020 [32] for continuity. The results were qualitatively consistent. In the marker-reordered analysis we count 45 clonal components whereas previously there were 46. Among the original 46, 44 are identical, while two differ: two samples constituting one clonal component were no longer considered clonal; another clonal component had one additional sample; see https://github.com/aimeertaylor/ColombianBarcode. Due to these minor differences, some original clonal component labels are offset by one.

## Supporting information

**S1 Fig. Principal component analysis of *P. falciparum* genomes obtained in different geographical regions of South America.** (A) Parasite samples from the East of the Andes mountain range (Guyana and Venezuela) are separated by principal component 1 from parasites from the West (Colombia and Ecuador). (B) Parasite sample SPT26239 is separated from other Ecuador and Colombia samples by principal component 2. (C) Parasite samples from Colombia and Ecuador are separated from Pf048 by principal component 3.
(EPS)

**S2 Fig. Network analysis of pairwise relatedness visualized using incremental thresholds of IBD, ranging from 0.1 to 0.9, and colored by country included in the analysis from the sampling period 2013–2017.**
(EPS)

**S3 Fig. ADMIXTURE groups shown in Fig 3 and sample equivalency to previously reported groups using STRUCTURE.** (A) Elbow method to identify the optimal number of ancestral populations from ADMIXTURE (K) based on cross-validation error. Red line indicates K = 5 ancestral populations selected following best practices. (B) ADMIXTURE

results highlighting the three ancestral populations identified in Knudson *et al.* 2020 with STRUCTURE [19].
(EPS)

**S4 Fig. ADMIXTURE groups using altered clonal cluster frequencies.** (A) Elbow method to identify the optimal number of ancestral populations from ADMIXTURE (K) based on cross-validation error. (B) ADMIXTURE analysis of the Colombia-Venezuela samples with altered cluster frequencies. Cluster C was decreased from a frequency of 0.09 to 0.016 and cluster E1 was increased from a frequency of 0.015 to 0.06. This analysis best supports a division into five groups ($K = 5$), of which only two fully match with the original analysis. As expected, the cluster that increased in frequency (E1) is fully ascribed to a single ADMIXTURE group (along with E2) whereas the cluster that decreased in frequency (C) is described as "admixed".
(EPS)

**S5 Fig. IBD and genetic distance calculations are highly correlated within the Pacific Coast sample set at the cluster level (Pearson's *r* = -0.96).** Mean genetic distance was calculated as the proportion of dissimilar calls within a set of 28,278 high-confidence SNPs. Both mean pairwise genetic distance and mean pairwise IBD were averaged over all inter-sample comparisons between each pair of clusters. Estimates for pairwise comparisons incorporating either of the two Venezuelan samples diverge to a greater extent, perhaps reflecting the inaccuracy of IBD estimation for samples originating outside the focal population. This is expected since population-level allele frequency estimates are required for IBD calculations, and these likely differ for the samples' true population of origin.
(TIF)

**S6 Fig. IBD analysis performs robustly across samples with different genotyping approaches.** (A) In order to minimize cliques within components in the extended analysis, one clonal component that contained six cliques was broken down into three clusters: the largest containing two cliques, the other two continuing a single clique each. (B) IBD point estimates between clonal clusters were obtained with both full WGS data (x-axis) and using information from only 250 sites (y-axis). These estimates are highly correlated (Pearson's r = 0.93). Values are calculated as the mean IBD of all between-cluster pairwise comparisons. (C) Exact sampling locations for clonal clusters depicted in Fig 4.
(TIF)

**S7 Fig. Network plot of clonal clusters (0.99 IBD) harboring distinct haplotypes for four main genes involved in antimalarial drug resistance.** Top panels correspond to *crt* (left) and *mdr1*, involved in resistance to chloroquine. Bottom panels are for *dhfr* (left) and *dhps* (right), involved in resistance to pyrimethamine and sulfadoxine, respectively. No clusters show signs of *de novo* mutations at known resistance alleles.
(EPS)

**S8 Fig. Plot displaying haplotypes surrounding the gene amino acid transporter 1 (aat1).** Samples carrying ancestral alleles isolated in Guapi (SPT26239) and two samples originating in Venezuela are shown at the top and bottom, respectively.
(EPS)

**S9 Fig. Mean population-wide pairwise IBD between clusters with origin in Ecuador and Colombia (2013–2017) calculated within 50-kb windows.** Each cluster was represented by the sample member with the most complete genome coverage. The dashed horizontal line marks the genome-wide 95th percentile. Windows falling above this threshold are marked

with red points. Vertical lines mark known resistance loci: *dhfr*, *mdr1*, *aat1*, *crt*, *dhps* and *kelch13*.
(TIF)

**S10 Fig.** (A) WGS samples retained for downstream analysis varied in coverage with 30–97% of the genome being covered at 5x read depth. Median 5x coverage was 83%. (B) The number of informative sites reported by hmmIBD (y-axis) varied with sample coverage (x-axis). (C) Samples with 30% of the genome covered at 5x read depth performed comparably in hmmIBD to samples with high coverage. Depicted are the 23 samples within clonal cluster E3, plotted based on their genomic coverage (x-axis). Fractional IBD calculations were made for each E3 sample to 12 other samples that varied in relatedness (y-axis) and genome coverage (color). Naively, each comparator sample should have the exact same fractional IBD to all E3 samples as E3 sample genomes are clonal replicates. Overall, replicability of IBD estimates across E3 samples is high, with a slight deflation when coverage is low. (D) When samples with <30% of the genome at 5x were included in the analysis, artifacts and biases appeared in the data. These included spurious clique formation in the igraph analysis and an increase in the standard deviation among fractional IBD estimates made to members of the same clonal cluster.
(PDF)

**S1 Table. IDs and metadata for all WGS samples used in study.**
(XLSX)

**S2 Table. Information on clonal clusters defined in study.**
(XLSX)

**S3 Table. WGS sample accession numbers for Colombia.**
(XLSX)

**S4 Table. Recoded genotype information from Echeverry, *et al* 2013 [17].**
(XLSX)

**S5 Table. Summary of data set types and analyses involving them.**
(XLSX)

## Acknowledgments

From Colombia we thank the Guapi communities, the Secretaría Municipal de Salud del Cauca and Secretaría Departamental de Salud del Cauca, as well as the VeuPathDB outreach team for assistance in the GoldenGate decoding SNP process, with the Colombian samples. We thank Marco Galardini for bioinformatics support and helpful comments. In particular, we thank the communities in Esmeraldas and San Lorenzo and the Health districts (especially Drs. Javier Obando, César Diaz and Julio Valencia).

## Author Contributions

**Conceptualization:** Manuela Carrasquilla, Angela M. Early, Aimee R. Taylor, Angélica Knudson Ospina, Julian C. Rayner, Fabián E. Sáenz, Daniel E. Neafsey, Vladimir Corredor.

**Data curation:** Manuela Carrasquilla, Angela M. Early, Diego F. Echeverry, Vladimir Corredor.

**Formal analysis:** Manuela Carrasquilla, Angela M. Early, Aimee R. Taylor.

**Funding acquisition:** Timothy J. C. Anderson, Caroline O. Buckee, Julian C. Rayner, Fabián E. Sáenz, Daniel E. Neafsey, Vladimir Corredor.

**Investigation:** Manuela Carrasquilla, Angela M. Early, Aimee R. Taylor, Angélica Knudson Ospina, Samanda Aponte, Pablo Cárdenas.

**Methodology:** Manuela Carrasquilla, Angela M. Early, Aimee R. Taylor.

**Project administration:** Samanda Aponte, Daniel E. Neafsey, Vladimir Corredor.

**Resources:** Timothy J. C. Anderson, Elvira Mancilla, Samanda Aponte, Julian C. Rayner, Fabián E. Sáenz, Daniel E. Neafsey, Vladimir Corredor.

**Supervision:** Timothy J. C. Anderson, Caroline O. Buckee, Daniel E. Neafsey, Vladimir Corredor.

**Validation:** Manuela Carrasquilla, Angela M. Early.

**Visualization:** Manuela Carrasquilla, Angela M. Early.

**Writing – original draft:** Manuela Carrasquilla, Angela M. Early.

**Writing – review & editing:** Manuela Carrasquilla, Angela M. Early, Aimee R. Taylor, Angélica Knudson Ospina, Diego F. Echeverry, Timothy J. C. Anderson, Elvira Mancilla, Samanda Aponte, Pablo Cárdenas, Caroline O. Buckee, Julian C. Rayner, Fabián E. Sáenz, Daniel E. Neafsey, Vladimir Corredor.

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
