## [Decision Letter · Decision Letter 0]

12 Aug 2022

Dear Dr Carrasquilla,

Thank you very much for submitting your manuscript "Resolving drug selection and migration in an inbred South American Plasmodium falciparum population with identity-by-descent analysis" for consideration at PLOS Pathogens. As with all papers reviewed by the journal, your manuscript was reviewed by members of the editorial board and by several independent reviewers. The reviewers appreciated the attention to an important topic. Based on the reviews, we are likely to accept this manuscript for publication, providing that you modify the manuscript according to the review recommendations.

Both reviewers have mentioned the low threshold for genome wide coverage of 30% however I note that Table S1 indicates that most of the samples have high genome coverage. Please provide an additional supplementary figure showing the distribution of coverage values, and comment on how robust the measurements of IBD are for low coverage samples.

Sincerely,

Alyssa Barry

Guest Editor

PLOS Pathogens

Kami Kim

Section Editor

PLOS Pathogens

Kasturi Haldar

Editor-in-Chief

PLOS Pathogens

orcid.org/0000-0001-5065-158X

Michael Malim

Editor-in-Chief

PLOS Pathogens

orcid.org/0000-0002-7699-2064

Both reviewers have mentioned the low threshold for genome wide coverage of 30% however I note that Table S1 indicates that most of the samples have high genome coverage. Please provide an additional supplementary figure showing the distribution of coverage values, and comment on how robust the measurements of IBD are for low coverage samples.

Reviewer Comments (if any, and for reference):

Reviewer's Responses to Questions

**Part I - Summary**

Reviewer #1: (No Response)

Reviewer #2: In this article, Carrasquilla and Early et al. present whole genome sequence data from 166 P. falciparum isolates collected predominantly from Colombia and Ecuador and estimate parasite relatedness within and between countries based on identity-by-descent. They also examine the prevalence of drug resistance mutations and their corresponding haplotypes to understand the selection for resistance in these populations. The sequencing data will be a valuable contribution to the field, although the analyses and conclusions are somewhat confirmatory, and in a few instances, overstated. Specific comments and questions are noted below.

**Part II – Major Issues: Key Experiments Required for Acceptance**

Reviewer #1: (No Response)

Reviewer #2: 1) Genome coverage of >=30% is quite low. Can the authors discuss how this threshold was selected, as prior studies have often used much higher cutoffs for genome coverage? Did the authors examine whether there was any clustering based on missingness, particularly in the PCA and ADMIXTURE analyses?

2) Can the authors clarify whether the same ~16K SNPs used to evaluate IBD from the WGS data were also used in the PCA and ADMIXTURE analyses? I believe ADMIXTURE assumes independent sites, which would necessitate LD-pruning to be performed prior to analysis. Was this done? Also, if the purpose of the ADMIXTURE analysis is to identify the number of genetic subpopulations, then highly related isolates (i.e., members of the same clonal lineage) would have to be removed, including only one member from each clonal cluster in the analysis. This would also negate the need to test different cluster frequencies. If the purpose of the analysis was to determine if ADMIXTURE would identify the same clonal lineages identified using IBD estimates, then does it make sense to select K=5? The value of K was selected using the “elbow method’ based on the CV error plot (which is standard), but by design based on the value of K chosen for the analysis, the ADMIXTURE analysis would not identify all of the same clonal lineages. I suspect if K had been set to 17, the same clonal lineages would have been identified.

3) The authors have not analyzed how effective population size has changed over time, and thus, would not have evidence to conclude that drug resistance emerged under current demographic conditions, as stated in the conclusions. Particularly for older first line antimalarials, this selection likely took place decades ago, during a time when effective population size may have been higher than it is in the current sample set.

**Part III – Minor Issues: Editorial and Data Presentation Modifications**

Reviewer #1: This article very clearly demonstrates the use of relatedness analysis to infer population clusters that may be related to transmission events but also how this can help identify regions of the genome that are under selection from interventions, mainly drugs used against the malaria parasite. As malaria elimination programs drive down the prevalence of the disease, such analysis set the scene for future applications by malaria genomic surveillance programs. There are however minor issue that could help the wider audience better comprehend the data and approaches applied.

1. The data is significantly heterogeneous, including both temporal and spatial data but with a variety of marker sets used for different analyses. While the authors have carefully identified and justified the use of these, it could remain challenging to follow for many, especially those being newly introduced into this area. A supplementary table summarising the new and previous data could be helpful. In this context, not only 166 isolates are analysed as indicated in the supplement.

2. The regional IBD is overrepresented by Columbia. At what level is IBD considered high and what statistics was applied for 0.36 between Colombia and Ecuador to be considered high? IBD is in the core of the analysis, and this has been done with whole genome data and various SNP sets. As genome coverage could be as low as 30% and these were compared with those having higher coverage, the sets of markers across the ~16000 that are common to pairs of isolates could be summarised in supplements. An extension of this heterogeneity is evident in the wide range used from SNP barcodes (12 to 248). The authors did indicate that they used the IBD CI to retain reliable IBD. What was the proportion of results retained from the low SNP numbers? This could inform the minimal number needed by those that would be applying this approach in resource limited settings.

3. A significant amount of masking was done to reduce heterozygous calls and therefore artificially assigning monogenomes. An indication of the number of such sites masked and if they indeed passed quality filters for sequencing will be informative. Complex infections remain a challenge in higher transmission regions and managing this with new methods, rather than discarding them could be helpful. Notwithstanding, approximating clonality to 1 in this analysis enabled more reliable IBD estimates but the unfiltered data could help in better appreciating complexity for these populations.

4. The contribution of regions of selective sweeps to overall pairwise IBD could be biased. As these are selective signatures, isolates are more likely to be in IBD in these regions due to selection even though they may not be from the same lineage. The authors could discuss how this affected genome-wide pairwise IBD. As the variants included between pairs did vary, this may add to the heterogeneity as some pairs of isolates will have larger proportions of sweep variants being analysed.

5. The igraph plots in supplementary figures 2 and 7 could be placed in separate boxes or borders added to distinguish between sub-plots

Reviewer #2: 4) The results section includes substantial amounts of data interpretation that would normally be included in the discussion section.

5) Minor: Although the number of polyclonal infections is relatively small, the predominant clone (if present) could be included in the analysis. As MOI is likely low in this setting (even if not equal to one), deconvolution of genomes using new tools may allow inclusion of genomes from these infections, rather than excluding them.

6) Minor: In the drug resistance literature, the notation of single, double, and triple mutant (as noted in Figure 5c) is often not used for genes other than dhfr and dhps, e.g., pfcrt. This made interpretation of the figure challenging.

PLOS authors have the option to publish the peer review history of their article (what does this mean?). If published, this will include your full peer review and any attached files.

Reviewer #1: No

Reviewer #2: No

Figure Files:

Data Requirements:

Reproducibility:

References:

---

## [Editor Report · Decision Letter 1]

9 Nov 2022

Dear Dr Carrasquilla,

We are pleased to inform you that your manuscript 'Resolving drug selection and migration in an inbred South American Plasmodium falciparum population with identity-by-descent analysis' has been provisionally accepted for publication in PLOS Pathogens.

Best regards,

Alyssa Barry

Guest Editor

PLOS Pathogens

Kami Kim

Section Editor

PLOS Pathogens

Kasturi Haldar

Editor-in-Chief

PLOS Pathogens

orcid.org/0000-0001-5065-158X

Michael Malim

Editor-in-Chief

PLOS Pathogens

orcid.org/0000-0002-7699-2064
---

## [Editor Report · Acceptance letter]

14 Dec 2022

Dear Dr Carrasquilla,

We are delighted to inform you that your manuscript, "Resolving drug selection and migration in an inbred South American Plasmodium falciparum population with identity-by-descent analysis," has been formally accepted for publication in PLOS Pathogens.

Best regards,

Kasturi Haldar

Editor-in-Chief

PLOS Pathogens

orcid.org/0000-0001-5065-158X

Michael Malim

Editor-in-Chief

PLOS Pathogens

orcid.org/0000-0002-7699-2064